# Significant Broad-Spectrum Antiviral Activity of Bi121 against Different Variants of SARS-CoV-2

**DOI:** 10.3390/v15061299

**Published:** 2023-05-31

**Authors:** Bobban Subhadra, Ragini Agrawal, Virender Kumar Pal, Agnes-Laurence Chenine, Jeffy George Mattathil, Amit Singh

**Affiliations:** 1Biom Pharmaceutical Corporation, 2203 Industrial Blvd, Sarasota, FL 34234, USA; 2Department of Microbiology and Cell Biology, Center for Infectious Disease Research, Indian Institute of Science (IISc), CV Raman Ave., Bengaluru 560012, Indiavirenderk@iisc.ac.in (V.K.P.); 3Integrated BioTherapeutics Inc., 4 Research Court, Suite 300, Rockville, MD 20850, USA; alchenine@gmail.com (A.-L.C.); jeffygeorge@gmail.com (J.G.M.)

**Keywords:** SARS-CoV-2, COVID-19, *Pelargonium sidoides*, Bi121, antiviral, Neoilludin B, intercalating agent

## Abstract

Severe acute respiratory syndrome coronavirus 2 (SARS-CoV-2) has so far infected 762 million people with over 6.9 million deaths worldwide. Broad-spectrum viral inhibitors that block the initial stages of infection by reducing virus binding and proliferation, thereby reducing disease severities, are still an unmet global medical need. We studied Bi121, which is a standardized polyphenolic-rich compound isolated from *Pelargonium sidoides*, against recombinant vesicular stomatitis virus (rVSV)-pseudotyped SARS-CoV-2S (mutations in the spike protein) of six different variants of SARS-CoV-2. Bi121 was effective at neutralizing all six rVSV-ΔG-SARS-CoV-2S variants. The antiviral activity of Bi121 was also assessed against SARS-CoV-2 variants (USA WA1/2020, Hongkong/VM20001061/2020, B.1.167.2 (Delta), and Omicron) in Vero cells and HEK-ACE2 cell lines using RT-qPCR and plaque assays. Bi121 showed significant antiviral activity against all the four SARS-CoV-2 variants tested, suggesting a broad-spectrum activity. Bi121 fractions generated using HPLC showed antiviral activity in three fractions out of eight against SARS-CoV-2. The dominant compound identified in all three fractions using LC/MS/MS analysis was Neoilludin B. In silico structural modeling studies with Neoilludin B showed that it has a novel RNA-intercalating activity toward RNA viruses. In silico findings and the antiviral activity of this compound against several SARS-CoV-2 variants support further evaluation as a potential treatment of COVID-19.

## 1. Introduction

The SARS-CoV-2 pandemic has infected over 762 million people with over 6.9 million deaths, markedly affecting human life and existence. In the US alone, over 1.12 million deaths have been reported as of March 2023 [1]. Since its first report, several SARS-CoV-2 lineages evolved in the last 24 to 30 months [2]; currently, the Omicron variant BA.2 is dominant around the world. The Omicron variant was declared as the fifth SARS-CoV-2 variant of concern (VOC), but researchers are monitoring other variants. Evolutionary viral mutations in spike and other proteins in highly evolved lineages may provide survival benefits to the virus to thwart the human immune system [3]. The Omicron variant carries 46 high-prevalence mutations specific to Omicron; twenty-three of these are localized within the spike (S) protein and the rest are localized to the other three viral structural proteins of the virus [4]. An Omicron variant XBB.1.16, which is a dominant variant in India, is now considered a variant of interest by the World Health Organization [5].

The FDA has authorized several antiviral medications to treat COVID-19 in people. Intravenous remdesivir (Veklury) was the first antiviral drug approved for use in adults and pediatric patients for the treatment of COVID-19. Remdesivir is an RNA polymerase inhibitor with a broad spectrum of antiviral activities against RNA viruses, especially in SARS-CoV, the Middle East respiratory syndrome (MERS), and SARS-CoV-2 [6]. Remdesivir reduced the risk of hospitalization or death by 87% compared with a placebo [7]. Oral ritonavir in combination with nirmatrelvir (Paxlovid), which was developed by Pfizer, was found to reduce the risk of hospitalization or death by 89% compared with a placebo if given early [8]. Molnupiravir (Lagevrio), an oral antiviral manufactured by Merck, was authorized by the FDA for the treatment of mild-to-moderate COVID-19. In people with mild-to-moderate COVID-19, molnupiravir reduced the risk of hospitalization and death by 30% [9]. Several monoclonal antibodies (bebtelovimab, sotrovimab, bamlanivimab/etesevimab, and casirivimab/imdevimab) are authorized to use through “emergency use authorization” (EUA) for the treatment of certain patients with COVID-19 by the FDA. Because of the prolific nature of the viral infectivity and evolutionary pressure from the vaccine-induced adaptive immune response and prospective antiviral treatments, there is a high probability of the emergence of more infectious SARS-CoV-2 variants with potential vaccine and antiviral drug resistance [10]. This suggests the need for antivirals with broad-spectrum neutralization activity toward various lineages of existing and future SARS-CoV-2 variants.

Several single-compound antivirals have shown themselves to be effective against SARS-CoV-2 in preclinical studies and are in clinical development [11,12]. However, these single-molecule-, single-target-based approaches may be challenging to implement effectively in rapidly evolving virus strains of SARS-CoV-2. Most of the current antiviral molecular targets are focused on the protein-based steric hindrance approach. The major protein-based targets are the receptor-binding domain (RBD) of the viral spike (S) protein to engage with angiotensin-converting enzyme 2 (ACE2) in host cells [13] or the RNA polymerase [14] or viral protease [15] necessary for replication.

Several antiviral compounds are in different stages of clinical trials, but broad-spectrum natural compounds that are effective at inhibiting coronavirus growth and replication with potentially new molecular targets can help in the fight against COVID-19. Many successful drug products are derived from medicinal herbs and then re-engineered to produce the active compound using synthetic chemistry [16].

A broad-spectrum viral inhibitor that targets various lineages of SARS-CoV-2 by blocking the initial stage of infection, including virus uptake and proliferation, may reduce the infectious viral load and, thereby, disease severity in patients. Further, a combination of antivirals that have different molecular targets may be critical to avoiding drug resistance and prolonging the therapeutic life cycle of the antiviral compound. For example, a combination of a protein-based viral target and a nucleic-acid-based target may reduce the probability of viral drug resistance evolution while delivering the intended benefit.

We isolated and studied Bi121, an aqueous polyphenolic-rich fraction from *Pelargonium sidoides* (PS). The aqueous ethanolic PS extract has been used as a traditional medicine for the treatment of various ailments for over a century [17]. A proprietary extract from PS roots known as EPs7630 or Umckaloabo [18] was evaluated in numerous clinical trials for safety and alleviation of symptoms associated with acute bronchitis and is licensed in Germany for the treatment of upper respiratory tract infections [19]. PS extract contains numerous metabolites and was reported to inhibit viruses associated with respiratory diseases, such as influenza viruses [20,21,22]; HIV [23]; and the herpes virus [24]. Very recently, EPs7630 was shown to have activity against SARS-CoV-2 [25]. We are exploring whether Bi121 has new molecular anti-viral targets that will add value to the current protein-based antiviral target as a combination therapy.

More than 30 clinical trials have been conducted with EPs7630 over the last 25 years (total study population > 10,500) in the treatment of acute respiratory tract infections. PS extract has excellent safety and tolerability in human clinical studies [17]. This corroborates the safety of Pelargonium-based compounds for human applications. Here, we investigated the antiviral activity of Bi121 against several strains of SARS-CoV-2. The data provide insight into the broad-spectrum antiviral activity of Bi121 against several SARS-CoV-2 variants and show preliminary insight into a novel RNA-intercalating activity toward RNA viruses.

## 2. Materials and Methods

### 2.1. Production and Standardization of Bi121

Crude PS extracts were generated from dried plant roots of PS sourced from South Africa. Extracts were prepared as described previously [23]. Roots were powdered by grinding with a ball mill and 100 g powdered roots were stirred in 600 mL water (ddH_2_O) for 24 h at 55 °C. This aqueous extraction was ultrasonicated (Sonicator 4000, Misonix Inc., Farmingdale, NY, USA), and the mixture was centrifuged to obtain a clear aqueous extract. All PS extract stock solutions were sterilized via filtration and stored at −20 °C until used for preliminary functional assays or polyphenolic enrichment.

### 2.2. Enrichment of Polyphenolic Rich Bi121

For polyphenolic enrichment, aqueous PS extract was thoroughly mixed with polyvinylpyrrolidone (PVPP, Sigma-Aldrich, #9003-39-8); the polyphenols were allowed to adsorb to PVPP at room temperature for 20 min. The mixture was filtered and washed three times with ddH_2_O. Polyphenols were eluted three times each with 0.5 mL 0.5 N NaOH. The enrichment of polyphenolic extracts and fractionation procedure was performed as previously described with some modifications.

### 2.3. Fractionation of Bi121 Polyphenol Fractions Using Reversed-Phase UHPLC

The fractionation of the polyphenol-enriched fraction was carried out using an Agilent AdvanceBio Column (2.7 µm, 2.1 × 250 mm) with solvent A (10 mM TEABC, pH 8.0) and an Agilent UHPLC 1290 system. The separation was performed by running a gradient of solvent B (10 mM TEABC, pH 8.0, 90% ACN) and solvent A (10 mM TEABC, pH 8.0) at the flow rate of 250 µL/min. The eluted fractions were collected in a 96-well plate using a 1260 series autosampler fraction collector based on the peaks at a UV wavelength between 214 nm and 280 nm. The 96-well plate eluted fractions were transferred to 1.5 mL tubes according to the retention time (12 min per fraction) for a total of 8 fractions. Six UHPLC runs were performed, and the eight fractions were pooled and further evaporated by using a speed vacuum (Thermo Scientific Savant, Waltham, MA, USA, ISS110).

### 2.4. Nanospray LC/MS/MS Analysis and Identification of Active Compounds

The LC/MS analyses of the active fractions were carried out using a Thermo Scientific Orbitrap Q Exactive Mass Spectrometer and a Thermo Dionex UltiMate 3000 RSLCnano System. Each active fraction was loaded onto a peptide trap cartridge at a flow rate of 5 μL/min. The trapped peptides were eluted onto a reversed-phase 20 cm C18 PicoFrit column (New Objective, Woburn, MA, USA) using a linear gradient of acetonitrile (3–36%) in 0.1% formic acid. The elution duration was 100 min at a flow rate of 0.3 μL/min. Eluted peptides from the PicoFrit column were ionized and sprayed into the mass spectrometer using a Nanospray Flex Ion Source (Thermo Scientific, Waltham, MA, USA, ES071) with a spray voltage of 1.6 kV and a capillary temperature of 250 °C. The Q Exactive instrument was operated in the data-dependent mode to automatically switch between full scan MS and MS/MS acquisition. Survey full scan MS spectra (*m*/*z* 150–600) were acquired in the Orbitrap with 70,000 resolutions (*m*/*z* 200) after the accumulation of ions to a 1 × 10^6^ target value based on predictive automatic gain control (AGC). Dynamic exclusion was set to 10 s. The 15 most intense multiply charged ions (z ≥ 1) were sequentially isolated and fragmented in the octupole collision cell via higher-energy collisional dissociation (HCD) using normalized HCD collision energy at 35% with an AGC target 1 × 10^5^ and a maximum injection time of 100 ms at 17,500 resolving power. MS Raw data files were analyzed using Compound Discoverer 3.3 software (Thermo Scientific, San Jose, CA, USA). MS1 peak area relative quantification was used for determining the relative abundance of each unique compound identified in each active fraction.

### 2.5. Cell Lines and Virus

Vero E6 and HEK 293T cells expressing human ACE2 cells (NR-52511, BEI Resources, NIAID, NIH, RRID: CVCL_A7UK) were cultured in complete media containing Dulbecco’s modified Eagle medium (#CC3004, Cell Clone, Genetix, New Delhi, India) with 10% CELLECT FBS GOLD (#2916754, MP Biomedicals, Santa Ana, CA, USA), 100 IU/mL Penicillin, 100 μg/mL Streptomycin, and 0.25 μg/mL Amphotericin-B (Antibiotic-Antimycotic Solution, #A5955, Sigma-Aldrich, St. Louis, MO, USA).

SARS-CoV-2 isolates (Hong Kong/VM20001061/2020, NR-52282; hCoV-19/USA/MD-HP05285/2021, NR-55671 (Delta Variant)) were obtained from BEI Resources, the NIAID, and the NIH and were propagated and tittered by using standard plaque assay in Vero E6 cells. The Omicron variant was isolated from the patient sample from the South Karnataka region, India, and was confirmed using S gene target failure PCR. All the experiments pertaining to live SARS-CoV-2 were performed in a viral biosafety level 3 laboratory at CIDR, IISc, Bangalore.

### 2.6. rVSV-ΔG-SARS-CoV-2 S Luciferase Pseudotype Assay and Cytotoxicity Assay

Pseudotyped viruses were generated based on a modification of a method described previously [26]. Briefly, the rVSV with the glycoprotein gene (G) deleted was used as the base platform for IBT’s pseudotype-based neutralization assays. The VSV-G glycoprotein was transiently expressed via transfection to produce virus particles in HEK293T cells. To create pseudotyped viruses, the VSV-G was substituted with SARS-CoV-2 spike protein (full-length spike protein lacking terminal eighteen amino acids of the cytoplasmic domain), and the resulting virus (rVSV-ΔG-SARS-CoV-2 S) could be handled at biosafety level 2 and expressed firefly luciferase. The infection efficiency could be measured by quantifying the luciferase activity by reading the relative light units (RLUs) on a luminometer. For the neutralization assay, 50 µL of each Bi121 dilution was mixed with 50 µL rVSV-ΔG-SARS-CoV-2 S variants in a 1:1 ratio for 1 h at 37 °C. All the media were removed from the 96-well plates, and the 100 µL mixtures were added in triplicate to Vero E6 cells for 24 h incubation at 37 °C; pseudotyped-virus-only and cell-only wells were included as controls. Luciferase activity was measured (Promega, Bright-Glo™ Luciferase Assay System) to determine the Bi121 IC_50_. Neutralization assays were validated using COVID-19+ rat serum. Neutralization was determined relative to the untreated virus controls. In the dose–response experiments, the concentration of the test article causing 50% neutralization (IC50) was determined. Data analyses were conducted using XLFit (4-parameter logistic model or sigmoidal dose–response model). Neutralization activity (IC_50_) of various dilutions of Bi121 against multiple rVSV-ΔG-SARS-CoV-2 S variants expressing spike protein mutations (Table 1) were tested in Vero E6 cells. Cytotoxicity of Bi121 (CC_50_) was assessed in parallel.

For cytotoxicity assays, Vero E6 cells were seeded in multiple black 96-well plates 24 h prior (day −1) at 5.00 × 10^4^ cells per well with EMEM medium (10% heat-inhibited fetal calf serum (FCS), 2 mM L-Glutamine, 1× Penicillin/Streptomycin (P/S)). On day 0, 2-fold dilutions of Bi121 were prepared. Media were removed from the 96-well plate and replaced with 50 µL of serum-free medium (Gibco, VP-SFM (1×)) supplemented with 2 mM L-Glutamine. A total of 50 µL of Bi121 dilutions were added in triplicate to Vero E6 cells and incubated at 37 °C for 24 h; wells with medium and cells alone were also included as controls. Luciferase activity was measured (Promega, CellTiter-Glo^®^ 2.0 Cell Viability Assay) and read using a Biotek Plate Reader.

### 2.7. Cytotoxicity Assay

Vero E6 and HEK-ACE2 cells were seeded in a 96-well cell culture dish at the densities of 10,000 cells/well for Vero E6 and 20,000 cells/well for HEK-ACE2 containing 100 μL complete media. Cells were incubated for 12 h at 37 °C in a humidified 5% CO_2_ incubator for adherence. After 12 h incubation, the media was replaced with fresh media, and cells were treated with various dilutions of Bi121 in triplicates. Untreated cells were considered a negative control, and DMSO-treated cells were considered vehicle controls. After the treatment, cells were incubated at 37 °C in humidified 5% CO_2_ incubator. Forty-eight hours post-treatment, 0.8 mg/mL MTT substrate (Thiazolyl Blue Tetrazolium Bromide, #M5655, Sigma) was added to each well and incubated at 37 °C in the dark until the formation of formazan crystals. Then, the culture media was carefully removed and blue formazan crystals were dissolved in 200 μL of DMSO. The purple color was read at 595 nm with a reference filter of 620 nm using a SpectraMax M3 plate reader (Molecular Devices).

### 2.8. Antiviral Effect of Bi121

Vero E6 cells were seeded in 24-well plates to reach complete confluency the next day. Cells were pretreated with a 1:40 dilution of Bi121 prepared in DMEM for 2 h and subsequently infected with SARS-CoV-2 at an MOI of 0.01 (Vero E6) or MOI of 0.1 (HEK-ACE2). Virus inoculum was removed after 1 h and the infection medium (DMEM with 2% FBS) was added to restore the initial dilution of Bi121. Cells were incubated at 37 °C in a humidified 5% CO_2_ incubator. After 48 h, the supernatant was processed for viral RNA isolation (mdi Viral isolation kit), followed by qRT-PCR, a plaque assay, and a TCID50 calculation.

### 2.9. Quantitative Real Time-PCR (qRT-PCR)

Total viral RNA was converted to cDNA as per the manufacturer’s protocol (Bio-rad iScript cDNA synthesis kit, #1708891). Reverse-transcribed cDNA was subjected to SYBR-based real-time PCR using iTaq Universal SYBR Green Supermix (Bio Rad #1725124). The viral copy number was estimated by generating a standard curve using SARS-CoV-2 genomic RNA of a known titer.

### 2.10. Plaque Assay

Plaque assays to measure infectious virus counts were performed as described before [27]. Briefly, Vero E6 cells were seeded in 6-well cell culture dishes to reach complete confluency the next day. Cells were washed once with 2 mL warm PBS and incubated with dilutions of cell culture supernatants in 200 μL complete DMEM for 1 h at 37 °C. The virus inoculum was then removed, and cells were overlaid with DMEM containing 2% FBS and 0.8% agarose (#A6013, Sigma-Aldrich, St. Louis, MO, USA). After 72 h incubation, cells were fixed with 4% formalin, and plaques were visualized via crystal violet staining (#C6158, Sigma-Aldrich, St. Louis, MO, USA).

### 2.11. Tissue Culture Infectious Dose 50 (TCID_50_)

Vero E6 cells were seeded in a 96-well cell culture dish to reach complete confluency the next day. Cells were incubated with dilutions of cell culture supernatant and prepared in an infection medium (DMEM + 2% FBS), with each condition having four biological replicates. Plates were incubated for 48 h, and the presence or absence of cytopathic effects was recorded. TCID_50_ was estimated using methods described by Reed and Muench [28].

### 2.12. Time of Addition Assay

Vero E6 was seeded in 24-well plates to reach complete confluency the next day and incubated overnight. The next day, Bi121 at the dilution of 1:40 was added to the cells either 2 h prior to infection (−2 h) and was present throughout infection (ON), or 2 h (+2 h) and 8 h (+8 h) post-infection. Then, cells were infected with 0.01 MOI SARS-CoV-2. The culture supernatants were collected at 48 hpi, and the viral load was calculated under different exposure conditions using RT-qPCR.

### 2.13. Target Prediction

Both 2D and 3D chemical structures of the compound Neoilludin B were obtained from the PubChem server [29]. The molecule was cross-referenced with the Therapeutic Target Database (TTD; https://db.idrblab.org/ttd/, accessed on 20 July 2022) and Pharmacogenomics Knowledge Base (PharmGKB; https://www.pharmgkb.org/, accessed on 20 July 2022), with “SARS-CoV-2” and “anti-viral” as the search terms. The target protein information was cross-referenced with known inhibitors and no similarity was found. Further, the compound structures were submitted to different servers for target component analysis. Servers such as the Swiss Target Prediction database [30], PharmMapper database [31], TargetHunter [32], ChemMapper [33], and ReverseScreen3D [34] were used to predict the plausible Neoilludin B protein targets and the top hits were further subjected to a homology search through the NCBI BLAST [35] server against the Homo sapiens and SARS-CoV-2 species.

### 2.14. Computational Validation

Molecular docking (MD) and MD simulation (MDS) validation pipelines were used to unravel potential targets of Neoilludin B. The 3D structures of SARS-CoV-2 helicase and RNA duplexes were obtained from a protein data bank [36] with PDB IDs 7NNG and 7KRP, respectively. The target structures were cleaned and prepared in Molecular Operating Environment (MOE) software version “MOE2022” updated on 6 July 2022 [37,38]. The water molecules with no hydrogen bindings with the core structure were deleted and errors regarding the bond orders and steric clashes were resolved using an automated wizard and the structure was optimized using default parameters. The standard whole-molecule docking was used to compare nonspecific scores as a baseline for the active site/major–minor groove docking. The alkylation potential of the docked structure was assessed using the proximity of cyclopropane with the nitrogen of nitrogenous bases or amino acid side chain residues.

The docked complexes were validated via MD simulations using the Schrödinger Desmond package [39,40]. The docked complexes were solvated in the default predefined TIP3P solvent model for both helicase–Neoilludin B and Neoilludin B–RNA/DNA duplex (the top 5 RNA duplex docked poses were superimposed on a single complex with no overlap between the ligands) complexes using the OPLS3e force field [41]. The orthorhombic box boundary with a 10 Å buffer space in all three planes was set. The salt concentration was set to 0.1 M with the access of Na^+^/Cl^−^ ions to keep the system neutral. The full system was loaded onto the Desmond simulation engine and set to run for 100 ns with default temperature and pressure settings (300 K/1 atm) in the OPSL3e forcefield. The simulation trajectories were analyzed for the root-mean-square deviation (RSMD) and fly-off times were documented based on visual inspection.

### 2.15. Statistical Analysis

All statistical analyses were performed using GraphPad Prism software version 9.0.0 (GraphPad Prism Software, Inc. San Diego, CA, USA). The data values are indicated as the mean ± S.D. For the statistical analysis, unpaired Student’s *t*-tests were used. Each error bar represents the standard deviation from the mean of two or three independent experiments. A *p*-value < 0.05 was considered significant.

## 3. Results

### 3.1. Bi121 Neutralized Pseudotyped-SARS-CoV-2 S (Spike Protein) Representing Various Viral Mutations

The antiviral activity of PS extracts against SARS-CoV-2 [25] was previously reported. Here, we studied the antiviral activity of Bi121 against SARS-CoV-2 using pseudotyped-SARS-CoV-2. Bi121 was tested for antiviral activity using recombinant vesicular stomatitis virus (rVSV)-pseudotyped SARS-CoV-2 S expressing the firefly luciferase. The neutralization activity of Bi121 (IC_50_) of various dilutions of Bi121 against multiple rVSV-ΔG-SARS-CoV-2 S variants expressing spike protein mutations (Table 1) was measured as luciferase activity to determine the number of viable cells.

Bi121 effectively neutralized multiple rVSV-ΔG-SARS-CoV-2 S variants expressing different mutations, with the Bi121 IC_50_ dilutions ranging from 183–1350 (Figure 1A–F). The Bi121 IC_50_ values against rVSV-ΔG-SARS-CoV-2 S WT and rVSV-ΔG-SARS-CoV-2 S Delta were 183 (Figure 1A) and 999.6 (Figure 1D), respectively. This preliminary screening showed the broad-spectrum neutralizing ability of Bi121 to rVSV-ΔG-SARS-CoV-2 with various spike protein mutations. This effect was shown with various SARS-CoV2 mutants, suggesting a broader inhibitory interaction.

### 3.2. Bi121 Neutralized Three Variants of SARS-CoV-2 in Two Different Cell Lines

As we observed the antiviral activity of Bi121 in pseudotyped-SARS-CoV-2, we next sought to determine the antiviral activity against different SARS-CoV-2 variants. To test the antiviral activity of Bi121 against SARS-CoV-2, Biom Pharmaceuticals partnered with the Center for Infectious Disease Research at the Indian Institute of Sciences (IISc., Bangalore, India). The cytotoxicity of Bi121 was measured in Vero E6 and HEK-ACE2 cell lines. Nontoxic concentrations were used for all the assays (Appendix A). The antiviral activity of Bi121 was assessed against three variants of SARS-CoV-2 using RT-qPCR, a plaque assay, and TCID_50_ estimation in two different cell lines (Vero E6 cells and HEK-ACE2). The three SARS-CoV-2 variants tested were USA WA1/2020, Hongkong/VM20001061/2020, and B.1.167.2 (Delta).

As shown in Figure 2, Bi121 showed significant activity toward all three SARS-CoV-2 variants tested in the two cell lines. In the Vero E6 and HEK-ACE2 cells, Bi121 at 1:40 dilution significantly reduced the viral replication (4–5 log reduction) compared with untreated using RT-qPCR, a plaque assay, and TCID_50_ estimation for USA-WA1/2020 and Hongkong/VM20001061/2020 strains (Figure 2). Similarly, Bi121 inhibited the B.1.167.2 (Delta) replication in Vero E6 and HEK-ACE2 cells by 4–5 log using RT-qPCR (Figure 2A,D), a plaque assay (Figure 2B,E), and a TCID_50_ assay (Figure 2C,F). Representative well images of the infected and treated cells also showed a reduction in the cytopathy effect induced by SARS-CoV-2 when treated with Bi121 (Appendix A). Our results demonstrated that Bi121 could significantly inhibit three different strains of SARS-CoV-2.

### 3.3. The Anti-SARS-CoV-2 Activity of Bi121 against Omicron Strain

Next, we sought to determine the antiviral activity of Bi121 against the Omicron strain. The Omicron strain was the dominant strain of the virus circulating in the US and the rest of the world during this study period; therefore, we tested the activity of Bi121 against the Omicron strain isolated at IISc, Bengaluru, India. Similar to the activity against other strains of SARS-CoV-2, Bi121 significantly reduced (5 log reduction) the viral copies in Vero E6 cells in the RT-PCR assays, reaffirming the broad-spectrum activity of Bi121 against SARS-CoV-2 (Figure 3). We observed significant antiviral activity of Bi121 against pseudotyped SARS-CoV-2 and different strains of SARS-CoV-2; therefore, we sought to undertake a more detailed characterization of the Bi121 fraction and its mechanism of antiviral activity.

### 3.4. Bi121 Interfered with the Early Stages of SARS-CoV-2 Infection

To determine at what stage of the viral life cycle the compound Bi121 imparted its antiviral activity, we performed a time of addition experiment. Vero E6 cells were either pre-treated with Bi121 (1:40 dilution) or left untreated for 2 h and subsequently infected with SARS-CoV-2 for 1 h. After 48 h post-infection, the medium supernatants were harvested and processed for RNA isolation and RT-qPCR. The schematic of the experiment is shown in Figure 4A. Bi121, when added 2 h before and after the infection, showed an inhibitory effect (*p* = 0.0011) (Figure 4B), whereas Bi121, when added 8 h after the infection, showed a less pronounced inhibitory effect (*p* = 0.0027) (Figure 4B). This suggests that Bi121 may interfere in the early steps of SARS-CoV-2 entry and replication. The inhibitory activity of Bi121 may derive from multivalent interactions between the polyphenolic molecules to bind tightly to the S-protein of SARS-CoV-2 to prevent the initiation of viral cell entry and replication. This effect was shown with various SARS-CoV-2 strains and multiple mutants, suggesting a broader inhibitory action of Bi121.

### 3.5. Antiviral Activity of Various Fractionated Polyphenolic Compounds from Bi121

To elucidate the anti-SARS-CoV-2 activity of Bi121, we fractionated the polyphenol-rich Bi121 into nine fractions (F0–F8) using HPLC for molecular characterization and to identify potential anti-SARS-CoV-2 compounds. The PVPP-enriched polyphenol and flavonoid fractions from 8 mL of Bi121 extract were prepared via enrichment using 0.2 g of PVPP40 (polyvinylpyrrolidone with an average molecular weight of 40,000). The fractionation of polyphenol enriched fraction was carried out using an Agilent AdvanceBio Column (2.7 μm, 2.1 × 250 mm) with solvent A (10 mM TEABC, pH 8.0) and an Agilent UHPLC 1290 system. The separation was performed by running a gradient of solvent B (10 mM TEABC, pH 8.0, 90% ACN) and solvent A (10 mM TEABC, pH 8.0) at the flow rate of 250 μL/m. The eluted fractions were collected according to the retention time (12 min per fraction) and a total of nine fractions were collected and evaporated by using a speed vacuum. The HPLC profile of the polyphenolic-rich Bi121 is shown in Figure 5.

The antiviral activity of various collected fractions, along with Bi121, was studied using an MTT assay. Briefly, Vero E6 cells were pretreated with an increasing concentration of each fraction (F0–F8) for 2 h and subsequently infected with SARS-CoV-2 at an MOI of 0.01. A 48 h post-infection MTT assay was performed to determine the cell viability. The cell viability was calculated after normalizing the data with an uninfected control. The percentage of viral inhibition was calculated by normalizing the viability of treated cells with respect to the viability of untreated controls as follows: % viral inhibition = [(cell viability after treatment − cell viability of untreated)/cell viability of untreated] × 100. A minimum of 10% viral inhibition was set as the cut-off value to determine the potential antiviral fraction.

The viral inhibition assays showed that B121.1 and fractions F2, F3, and F5 had antiviral activity against SARS-CoV-2 (Figure 6), suggesting multiple compounds with activity against SARS-CoV-2. Fraction 5 showed 42% viral inhibition at 6.25 μg/mL; however, increasing the concentration resulted in lower viral inhibition. Polyphenolic compounds tend to form polymeric compounds, which may be the reason for the lower activity at higher concentrations.

### 3.6. Identification of Potential Polyphenol and Flavonoid Fractions from the Bi121 Extract Using LC/MS/MS

Previous studies showed that the compounds responsible for the antiviral activity in *P. sidoides* are sesquiterpenes. We undertook a detailed molecular identification of the compounds in Bi121 fractions (F2, F3, F5) using mass spectroscopy. The liquid chromatography–tandem mass spectrometry (LC/MS/MS) analysis of active fractions was carried out using a Thermo Scientific Orbitrap Q Exactive Mass Spectrometer and a Thermo Dionex UltiMate 3000 RSLCnano System. Nine fractions were generated based on the LC retention time and the biological activity analysis of the nine fractions was performed. The results indicated that three fractions, namely, F2, F3, and F5, showed anti-viral activity (Figure 6).

The three fractions were analyzed via LC/MS/MS using a Q Exactive Orbitrap mass spectrometer and the compounds were identified. The MS Raw data files were analyzed by using Compound Discoverer 3.3 software (Thermo Scientific, San Jose, CA, USA). The MS1 peak area relative quantification was used for determining the relative abundance of each unique compound identified in each active fraction. One of the dominant compounds, namely, a 298.33 Da small molecule designated as Bi121.2, was detected in all three active fractions. Using structural analysis and a compound library comparison, Bi121.2 was putatively identified as Neoilludin B. Table 2 shows the presence of the active compound Bi121.2 and observed antiviral activity.

### 3.7. Target Prediction and Computational Target Validation

Swiss target prediction, TTD, and PharmGKB servers yielded no significant hits owing to a less characterized class of sesquiterpene chemicals. There are only a few reports describing similar molecules; however, their mechanism of action is mostly attributed to DNA methylation potential due to the presence of a cyclopropane sub-moiety [41,42]. The top 10 hits of PharmMapper were analyzed and found Zika virus NS3 helicase (PDB ID 6ADW) bound to 4-(2-Hydroxyethyl)-1-Piperazine Ethanesulfonic acid, which shares similar pharmacophore features with Neoilludin B. The TargetHuntersuggested a similarity to a ring extended nucleoside analogue inhibition of HIV helicase [43]. Furthermore, the docking scores of Neoilludin B with SARS-CoV-2 helicase (PDB ID 7NNG) were high enough (ΔG = −57.0163) to corroborate this fact (Figure 7).

Additionally, the MD simulations showed a lot of hinge-like movement with the helicase target protein with high RMSD and an early fly-off (~15 ns) of Neoilludin B, which was not due to a lack of binding energy but significant variations in binding pockets of the receptor. It is possible that in the presence of helicase interacting with SARS-CoV-2 NSPs and the RNA duplex substrate, the receptor complex will be more stable and effectively blocked by Neoilludin B. The ChemMapper server output pointed to a similar molecule being intercalating agents based on previous reports [44,45,46]. However, there was no significant docking observed with DNA. Conversely, RNA duplex docking showed multiple anchoring points for Neoilludin B (ΔG = −52.0163), specifically with adjacent A and U in the major groove (Figure 8 and Figure 9).

During the early phase of the MD simulation at ~30 ns, the Neoilludin B molecule “slid” between A–U and U–T stacked bases and formed a stable intercalated RNA duplex complex with a restricted horizontal pendulum-like movement for the rest of the simulation (Figure 8 and Figure 9). It is well known that the A–U bond is stronger than the A–T bond [47], while the Neoilludin B showed a strong pre-intercalation docking interacting with the uracil pyrimidine nitrogen, forming a cavity for the docking of the indene substructure, which acted as a gateway for the stronger intercalation during the simulation (Figure 8 and Figure 9). Further experimental wet validations are required to confirm the mode of action of Neoilludin B against the SARS-CoV-2 RNA duplex; however, these findings open new venues for targeted drug discovery, as there are reports of other RNA-duplex-specific intercalating agents [48].

Our computational analysis strongly suggests that Neoilludin B is an RNA-duplex-specific intercalating agent and therefore has the possibility of broad-spectrum antiviral activity against multiple RNA viruses with an RNA duplex as a part of their life cycle, which is absent in the host normal central dogma and the ribosomal RNA is heavily complexed with protein molecules. Furthermore, based on the strong docking scores, the binding pharmacophore can be used as a preliminary hit to search for more potent inhibitors.

## 4. Discussion

The last few years have seen the emergence of three novel human coronaviruses. Human coronaviruses normally cause the common cold in healthy people, but SARS-CoV-2 caused severe infections in human populations with an alarming global death toll of 7 million. To address the public health emergency of COVID-19 and emerging new SARS-CoV-2 variants and to maximize pandemic preparedness, broad-spectrum antivirals are required. A broad-spectrum antiviral against SARS-CoV-2 is a global need, as new virus lineages with more immune-thwarting mutations are evolving [49]. Adaptive mutations in the SARS-CoV-2 genome can increase the pathological potential, thereby making drug and vaccine development difficult [49]. Different variants of SARS-CoV-2 may arise in the future and broad-spectrum antiviral compounds can help with controlling viral binding and replication. Our studies showed that Bi121 with significant activity toward all the SARS-CoV-2 strains tested, potentially by interfering in the early steps of SARS-CoV-2 entry and replication. PS has several active compounds that are known to inhibit viruses [20,21,22,24]. EPs7630, which is a PS root extract, was found to be more efficacious than a placebo when given to patients with acute bronchitis [50]. Several compounds, such as sesquiterpenes, polyflavanoids, benzopyranones, and polymeric prodelphinidins, from PS root extract are responsible for the reported antiviral activity in *P. sidoides*. The main antiviral activity of PS extract is mediated by inhibiting the attachment and entry of viruses, representing early virus infection [21,51,52]. Similar to these observations, we found that a part of the antiviral activity of Bi121 against SARS-CoV-2 was based on virus entry inhibition. Our computational studies also showed that Bi121.2 Neoilludin B showed a strong RNA intercalating activity. We think Bi121.2 has multiple targets against SARS-CoV-2 and we are currently elucidating the detailed mechanism of action of Bi121.2.

An increased appearance of several viral infections in different parts of the world, either due to climate change, deforestation, or increased population densities, point to the development of effective broad-spectrum antiviral drugs. Bi121 has the potential to become a successful therapeutic because of its strong antiviral activity without cytotoxic effects and is a thermally stable compound. There is significant safety information [17,18,19,20,21,22] on the use of Pelargonium-based actives for human respiratory applications that warrant fast-tracked clinical development.

The nasal epithelium is where most respiratory viral infections, including SARS-CoV-2, infect the host [10]. The development of Bi121 in a nasal liquid dosage as a preventative treatment on nasal tissue may reduce viral infections. We are currently exploring the development of Bi121 as a therapeutic nasal spray for the first line of defense against SARS-CoV-2 infection. The SARS-CoV-2 infection in humans begins with the attachment and entry into the respiratory epithelium via the angiotensin-converting enzyme 2 (ACE2) receptor. Our antiviral studies with Bi121 suggest that Bi121 may interfere in the early steps of SARS-CoV-2 entry and replication with a novel mechanism of action. This effect was observed with various SARS-CoV-2 strains tested, suggesting a broader inhibitory action of Bi121. Through this study, we identified the dominant compound, namely, a 298.33 Da small molecule (Neoilludin B) designated as Bi121.2. Bi121.2 is a novel antiviral with a novel mechanism of action and, therefore, a high potential for use against COVID-19. Antibacterial activity of Neoilludins against *Bacillus subtilis* and *Staphylococcus aureus* was reported [53]. This was the first reported observation of the antiviral activity of Neoilludin B. Although, we identified the molecular entity as Neoilludin B, we do not know the exact chiral nature of the compound. Compounds such as Neoilludin B and Neoilludin C exist in nature as a mix of several related chiral structures [54]. Therefore, we are chemically synthesizing Neoilludin B and related analog compounds to elucidate the structural variations that pertain to the antiviral activity. Further, we are currently studying the activity of purified Neoilludin B against multiple RNA viruses. Our in silico analysis and computation models suggest that Neoilludin B is an RNA-duplex-specific intercalating agent and, therefore, has the possibility of broad-spectrum antiviral activity against multiple RNA viruses. The unique mechanism of action of this compound is expected to synergize with other leading anti-SARS-CoV-2 treatments, such as Molnupiravir and Nirmatrelvir. Infectious RNA viruses, such as Ebola and Marburg virus, cause serious lethal hemorrhage diseases [11,55,56] that have the potential to be pandemic; therefore, new drug candidates that are effective against RNA viruses are a global need. Further studies on the antiviral activity and mechanism of activity of Neoilludin B and its variants against these viruses can provide valuable anti-viral drug candidates to human disease prevention.

## 5. Summary

In summary, we demonstrated that Bi121 significantly inhibited various strains of SARS-CoV-2 in the two cell lines tested. This inhibition was also observed with the rVSV-ΔG-SARS-CoV-2 system that represents spike protein mutations of various SARS-CoV-2 lineages. Moreover, Bi121 used at concentrations that inhibited the virus did not display toxicity to the target cells. Bi121 pretreatment before infection showed an inhibitory effect, suggesting both treatment and prophylactic effects against SARS-CoV-2. The computational analysis strongly suggests Neoilludin B as an RNA-intercalating agent and, therefore, has the possibility of broad-spectrum antiviral activity against RNA viruses.

## Figures and Tables

**Figure 1 viruses-15-01299-f001:**
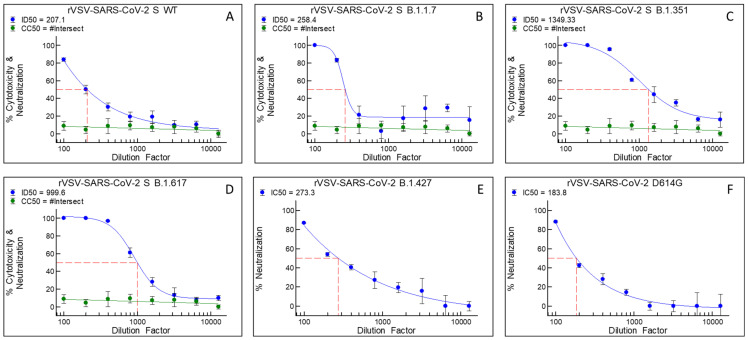
Cytotoxicity (**A**–**D**) and neutralization activity of Bi121 (**A**–**F**) against six rVSV-ΔG-SARS-CoV-2 S variants: (**A**) rVSV-ΔG-SARS-CoV-2 S WT, (**B**) rVSV-ΔG-SARS-CoV-2 S B.1.1.7, (**C**) rVSV-ΔG-SARS-CoV-2 S B.1.351, (**D**) rVSV-ΔG-SARS-CoV-2 S B.1.617, (**E**) rVSV-ΔG-SARS-CoV-2 S B.1.427, and (**F**) rVSV-ΔG-SARS-CoV-2 S D614G. Cytotoxicity is shown in green and was not assessed for rVSV-ΔG-SARS-CoV-2 S B.1.427 and rVSV-ΔG-SARS-CoV-2 S D614G. Neutralization is shown in blue.

**Figure 2 viruses-15-01299-f002:**
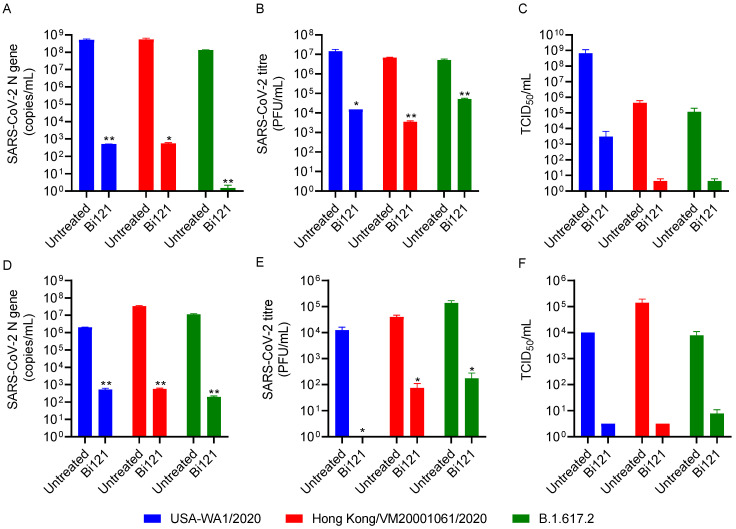
Anti-SARS-CoV-2 activity of Bi121: The antiviral activity of Bi121 was assessed in two different cell lines (Vero E6 (**A**–**C**) and HEK-ACE2 cells (**D**–**F**)) and against multiple strains of SARS-CoV-2 using RT-qPCR (**A**,**D**), a plaque assay (**B**,**E**), and TCID_50_ estimation (**C**,**F**). Vero E6 (**A**–**C**) and HEK-ACE2 (**D**–**F**) cells were pre-treated for 2 h with Bi121 (1:40 dilution) or left untreated and were subsequently infected with indicated SARS-CoV-2 strains at an MOI of 0.01 for Vero E6 and 0.1 for HEK-ACE2 cells for 1 h. The viral inoculum was washed, and the medium was replaced with fresh media (DMEM with 2% FBS) containing Bi121 or left untreated (UT). Post-infection at 48 h, the medium supernatant was harvested and processed for RNA isolation or serially diluted to perform the plaque assay and TCID_50_ estimation. SARS-CoV-2 RNA was isolated from 140 μL of supernatant and the levels of SARS-CoV-2-specific N-gene were assessed using RT-qPCR. The potential antiviral activity was assessed by comparing the compound-treated well with the untreated well and represented as the RNA copy numbers. (**A**,**D**) Infectious SARS-CoV-2 titers were assessed in Vero E6 cells using a standard plaque assay (**B**,**E**) and TCID_50_ estimation (**C**,**F**). The error bar represents the standard deviation from the mean of two independent experiments (*N* = 2), performed in duplicates. ** *p* < 0.001 and * *p* < 0.05 using an unpaired Student’s *t*-test.

**Figure 3 viruses-15-01299-f003:**
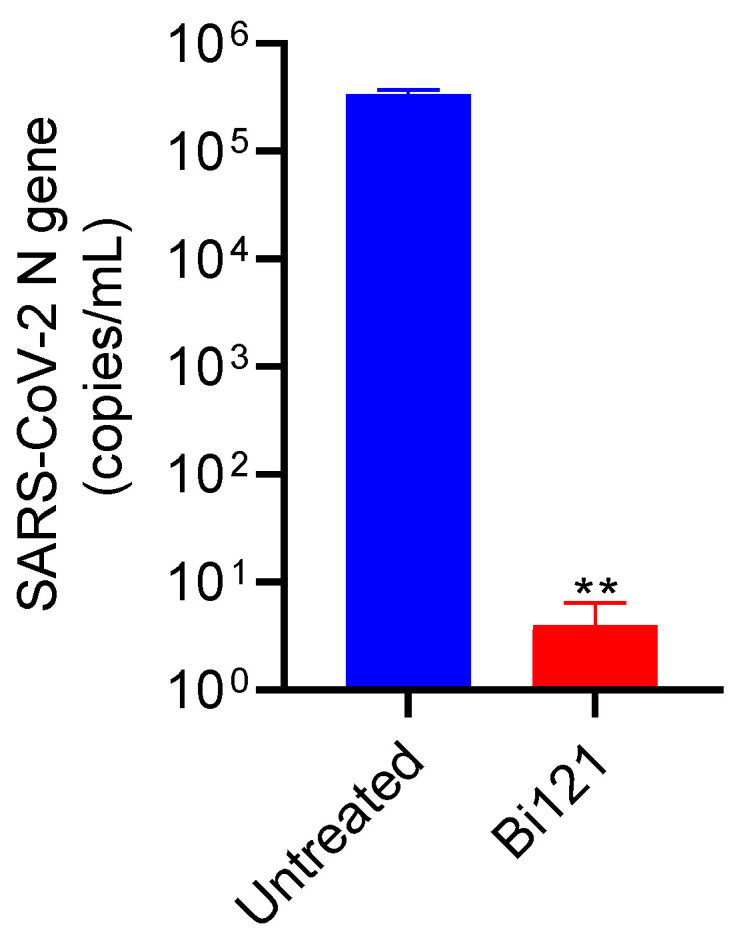
Anti-SARS-CoV-2 activity of Bi121 against the Omicron strain. The antiviral activity of Bi121 was assessed against the SARS-CoV-2 Omicron strain using RT-qPCR. Vero E6 cells were pre-treated for 2 h with Bi121 (1:40 dilution) or left untreated and were subsequently infected with the SARS-CoV-2 Omicron strain at an MOI of 0.5 for 1 h. After 1 h, the viral inoculum was washed, and the medium was supplemented with a fresh medium (DMEM with 2% FBS) containing Bi121 or left untreated (UT). Post-infection at 48 h, the medium supernatant was harvested and processed for RNA isolation. The SARS-CoV-2 RNA was isolated from 140 μL of supernatant and the levels of SARS-CoV-2-specific N-gene were assessed using RT-qPCR. The potential antiviral activity was assessed by comparing the compound-treated well with an untreated well and represented as the RNA copy numbers. Each bar represents the mean of two independent experiments (*N* = 2) performed in duplicates. ** *p* < 0.001 using an unpaired Student *t*-test.

**Figure 4 viruses-15-01299-f004:**
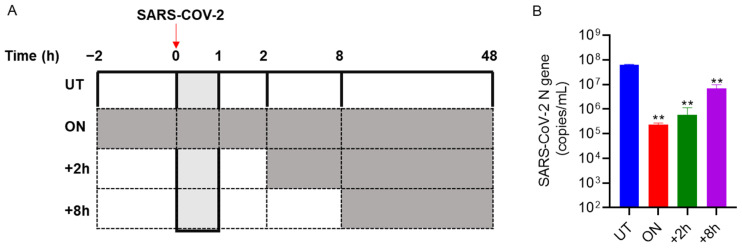
Early treatment with Bi121 inhibited SARS-CoV-2 replication. A time-of-addition assay was performed to identify the effect of Bi121 on SARS-CoV-2 replication. HEK-ACE2 cells were either pre-treated with Bi121 (1:40 dilution) or left untreated for 2 h and subsequently infected with 0.1 MOI of SARS-CoV-2 (Hongkong/VM20001061/2020) for 1 h. The viral inoculum was washed, and the medium was replaced with a fresh medium (DMEM with 2% FBS) containing Bi121 at indicated time points. Post-infection at 48 h, the medium supernatant was harvested and processed for RNA isolation. (**A**) Schematic of the time-of-addition experiment. Grey boxes represent the Bi121 treatment. (**B**) SARS-CoV-2 RNA was isolated, and the levels of SARS-CoV-2-specific N-gene were assessed using RT-qPCR. Potential antiviral activity was assessed by comparing the compound-treated wells with the untreated wells and represented as the RNA copy numbers. Each error bar represents the standard deviation from the mean of two independent experiments (*N* = 2) performed in duplicates. ** *p* < 0.001 using an unpaired Student’s *t*-test.

**Figure 5 viruses-15-01299-f005:**
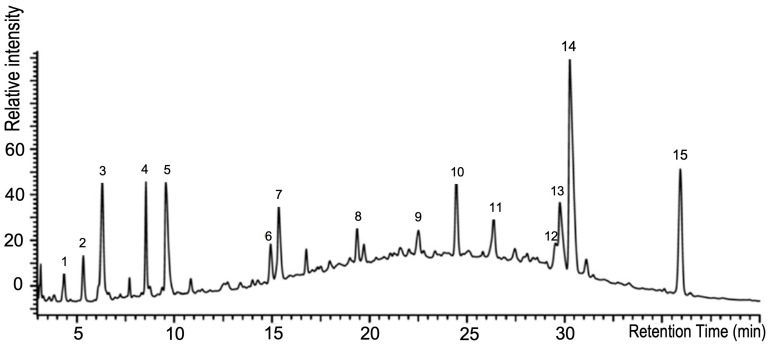
HPLC profile of the standardized polyphenolic-rich Bi121 with fifteen characteristic compound peaks.

**Figure 6 viruses-15-01299-f006:**
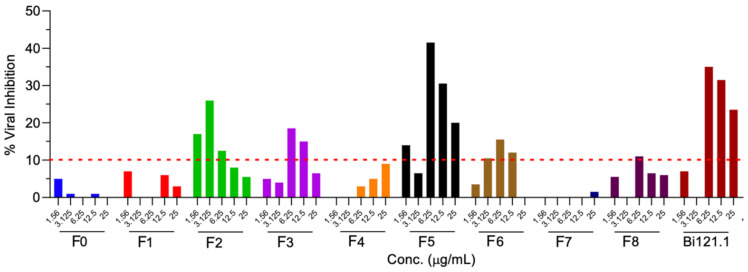
Anti-SARS-CoV-2 effect of different Bi121 fractions. Vero cells were pretreated with the indicated concentrations of compounds for 2 h and subsequently infected with SARS-CoV-2 at an MOI of 0.01. A 48 h post-infection MTT assay was performed to determine the cell viability. The cell viability was calculated after normalizing the data with uninfected control. The % viral inhibition was calculated by normalizing the viability of treated cells with respect to the viability of untreated controls as follows: % viral inhibition = [(cell viability after treatment − cell viability of untreated)/cell viability of untreated] × 100. A minimum of 10% viral inhibition was set as the cut-off value to determine the potential antiviral fraction.

**Figure 7 viruses-15-01299-f007:**
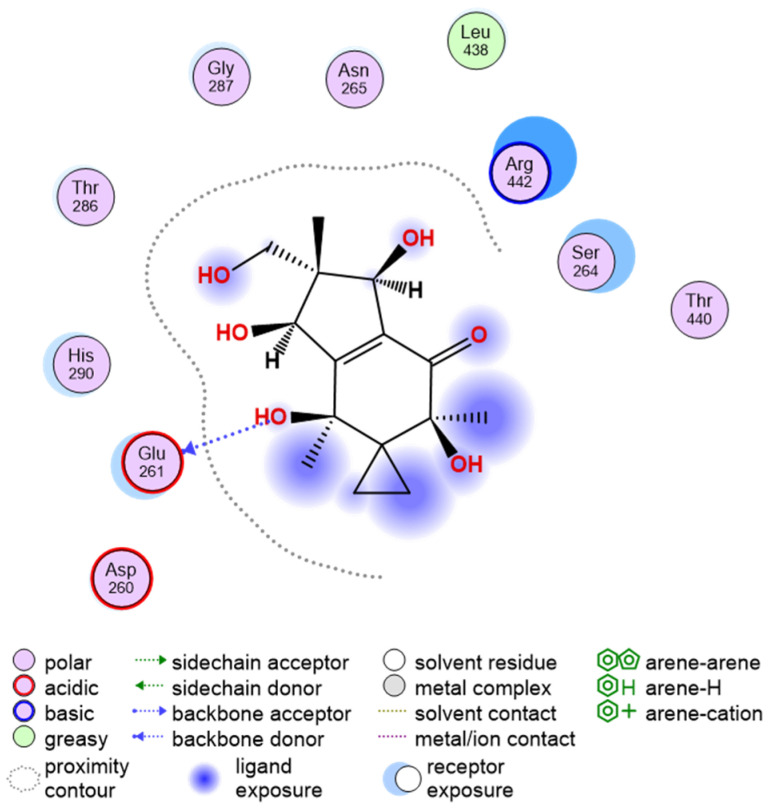
Schematic representation of the docking site of Neoilludin B with SARS-CoV-2 helicase. The whole molecule was involved in docking with a high score (ΔG = −57.0163) and the site of docking matched the reported site of nucleotide analog docking.

**Figure 8 viruses-15-01299-f008:**
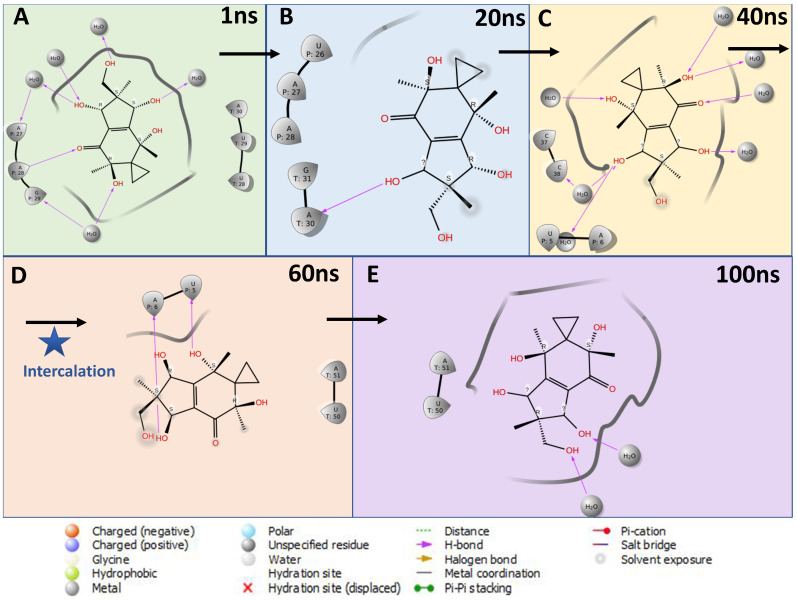
Schematic representation of the 2D interactions of Neoilludin B docked at adjacent U–A in a major groove at different stages of a 100 ns long MD simulation. (**A**) At 1 ns initial docking structure with a superficial interaction with the major groove was seen. (**B**) At 20 ns initial interactions with uridine at the 5th position was seen. (**C**) At almost half time (40 ns), the molecule started to loosen its interactions with the major groove and water molecules in favor of base stacking. (**D**) At 60 ns, the molecule was completely intercalated between adjacent U5 and A6 with no water molecule interactions. (**E**) At 100 ns, rest of the simulation, the molecule alternated by flipping and sliding in mirror U50 and A51 and back and multiple conformers suggested higher stabilization.

**Figure 9 viruses-15-01299-f009:**
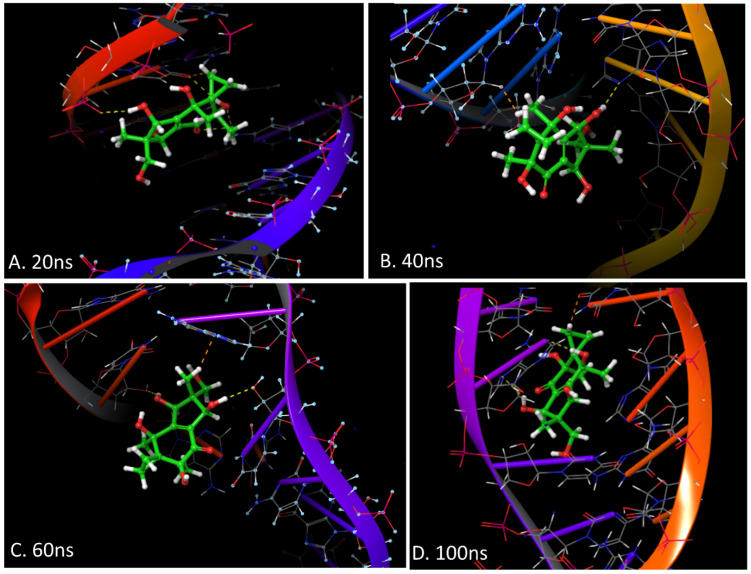
Schematic representation of the 3D interactions at different time points during 100 ns of the MD simulations of Neoilludin B docked at adjacent U–A in a major groove. (**A**) At 20 ns, the initial interactions with uridine in the 5th position of the leading RNA strand are seen with an initial interaction of pyrimidine nitrogen and the indene substructure of Neoilludin B. (**B**) At 40 ns, there was a seemingly unstable interaction, where the molecule was sort of hanging and losing major groove interactions of the original docking while squeezing in between stacked bases of adjacent U–A in the leading RNA strand. (**C**) At 60 ns, the molecule was completely intercalated between adjacent U5 and A6 with classic parallel nucleic acid base stacking “bent” to accommodate this intercalating agent. (**D**) At 100 ns, rest of the simulation molecule alternated by flipping and sliding in mirror U50 and A51 and back with high stability for Neoilludin B to fly off. Color code used in the figure are green: carbon, white: hydrogen, blue: nitrogen and red: oxygen.

**Table 1 viruses-15-01299-t001:** Specific mutations of VSV-pseudotyped spike protein that represent various strains of SARS-CoV-2.

rVSV-ΔG-SARS-CoV-2 S	Specific Mutations in the Spike Protein of Pseudotyped Strains
Wuhan wild type	-
B.1.1.7 (Alpha)	A570D, D614G, D1118H, delH69-V70, N501Y, P681H, S982A, T716I, delY144
B.1.351 (Beta)	K417N, E484K, N501Y, D614G, A701V
B.1.617 (Delta)	L452R, E484Q
D614G	D614G
B.1.427	L452R, D614G

**Table 2 viruses-15-01299-t002:** Antiviral activity of the 9 Bi121 fractions. Vero cells were pretreated with the indicated concentrations of compounds for 2 h and subsequently infected with SARS-CoV-2 at an MOI of 0.01. A 48 h post-infection MTT assay was performed to determine the cell viability. An MS analysis of the fractions showed the presence of Bi121.2 in the fractions F2, F3, and F5.

Fraction Identity	Putative Active Molecule	
	SARS-CoV-2 Antiviral Activity (10–20% Viral Inhibition)	Presence/Absence of 298.33 Da (Bi121.2) Small Molecule Using MS
Bi121 Fraction F0	No	No
Bi121 Fraction F1	No	No
Bi121 Fraction F2	Yes	Yes
Bi121 Fraction F3	Yes	Yes
Bi121 Fraction F4	No	No
Bi121 Fraction F5	Yes	Yes
Bi121 Fraction F6	No	No
Bi121 Fraction F7	No	No
Bi121 Fraction F8	No	No

## Data Availability

All data and materials are available upon request.

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
