# Peer review of "Significant Broad-Spectrum Antiviral Activity of Bi121 against Different Variants of SARS-CoV-2"

_viruses, 2023, doi:10.3390/v15061299_

Round 1

Reviewer 1 Report

The manuscript by Subhadra et al., entitled: Significant Broad Spectrum Antiviral activity of Bi121 Against Different Variants of SARS-CoV-2 demonstrates the cytotoxicity and antiviral efficacy of Bi121 in vitro against SARS-CoV-2. The in silico results also show the RNA-specific intercalating antiviral activity. Overall, the study is well-planed and the manuscript is very well organized. I have a few comments that may help improve the quality of the manuscript. Please find my comments below:

A few statements in the introductions as well as the discussion needs appropriate citations. Paragraph #2 and #3 in introduction for example. 

Authors should include a table with a safety index or therapeutic index 

Since Bi121 intercalates on the viral RNA, did authors consider pre-incubation with the virus in the time of the addition experiment? 

If possible, authors should perform a combination experiment of Bi121 with a positive control inhibitor like Remdesivir

Author Response

Responses to Reviewer#1 Comments/Recommendations

We thank the reviewer for their time/expertise for constructive critique and suggestions and have made the necessary changes to address the concerns/recommendations.

  1. A few statements in the introductions as well as the discussion needs appropriate citations. Paragraph #2 and #3 in introduction for example. 

Response: As suggested by the reviewer we have included appropriate citations both in introduction and discussion.

  1. Authors should include a table with a safety index or therapeutic index 

Response: We acknowledge the reviewer’s concern, but we do not have any data other than the toxicity data of Bi121, which we included in the manuscript. We are continuing the studies using the identified compound and is currently synthesizing the compound to study the compound and will be doing the safety index or therapeutic index.

  1. Since Bi121 intercalates on the viral RNA, did authors consider pre-incubation with the virus in the time of the addition experiment

Response:  The reviewer raises an important point. No, we did not pre-incubated the virus with the compound. The compound treatment was given only to the cells. We are synthesizing the compound to perform additional experiments to perform these studies.

If possible, authors should perform a combination experiment of Bi121 with a positive control inhibitor like Remdesivir

Response: We thank the reviewer for this suggestion. We will conduct combination study of Bi121 with other known antiviral/Remdesivir in the future and assess the relative merits of the combination. 

Reviewer 2 Report

The manuscript investigates a novel antiviral compound, Bi121, and its potential application against SARS-CoV-2. The results demonstrated that Bi121 effectively neutralized multiple rVSV-ΔG-SARS-CoV-2 S variants expressing different mutations, with IC50 values ranging from 183-1350. The antiviral activity of Bi121 against three SARS-CoV-2 variants was further assessed, showing significant activity towards all tested strains in Vero E6 and HEK-ACE2 cells. The authors then examined the antiviral activity of Bi121 against the Omicron strain, confirming a broad-spectrum activity of Bi121 against SARS-CoV-2. Time-of-drug-addition experiments suggested that Bi121 may interfere in the early steps of SARS-CoV-2 entry and replication, possibly through multivalent interactions with the S-protein. Bi121 was fractionated into nine fractions, and their antiviral activities were studied. Fractions F2, F3, and F5 showed antiviral activity, with a small molecule, Bi121.2, detected in all three active fractions. The molecule was putatively identified as Neoilludin B. Computational analysis strongly suggests that Neoilludin B may be an RNA duplex specific intercalating agent, offering potential for targeted drug discovery.

Although the topic is of significant interest, the manuscript requires substantial revisions to address concerns regarding methodology and presentation. Below, I have listed my minor and major comments, addressing each of the sections in the manuscript.

Minor comments:

1. Introduction:

a. Please ensure consistency in the terminology used for SARS-CoV-2 and its variants throughout the manuscript.

b. There are several instances of missing or incorrect citations. Please double-check your references and ensure all cited works are accurately represented in the reference list.

2. Materials and Methods:

a. Use the appropriate scientific nomenclature for the cell lines, viruses, and compounds mentioned in the manuscript.

b. Ensure that all chemicals, reagents, and equipment used in the study are listed along with the manufacturer's details.

3. Results:

a. The presentation of the results can be improved by incorporating figures and tables where appropriate. This would help readers visualize the data and better understand the key findings.

b. Make sure to include error bars or other measures of variability in any graphical representations of the data.

4. Discussion:

a. There are several instances of grammatical errors and awkward phrasing throughout the discussion. Please proofread the manuscript carefully and revise for clarity and fluency.

Major comments:

1. Introduction:

The introduction lacks a clear rationale for the study. Please provide a more comprehensive review of the current knowledge on the topic and highlight the gaps in the literature that your study aims to fill.

It is necessary to emphasize the significance of your study in the context of the ongoing COVID-19 pandemic and the need for new therapeutic approaches to tackle emerging variants.

The introduction needs to provide a more comprehensive review of the current knowledge on the topic, discussing key studies and their findings. Make sure to highlight the gaps in the literature that your study aims to fill and emphasize the significance of your study in the context of the ongoing COVID-19 pandemic.

2. Materials and Methods:

The description of the methods is insufficient to allow other researchers to reproduce the experiments. Please provide more detailed information on the experimental procedures, including any controls used, the number of replicates, and the specific statistical analyses performed.

The choice of cell lines and viral strains needs further justification. Explain the rationale behind their selection and discuss any potential biases that may arise from using these models.

The methods for the synthesis and purification of Bi121 and Neoilludin B should be described in more detail, including the specific protocols followed and any characterization performed to confirm their identity and purity.

The description of the methods needs elaboration to allow reproducibility. Provide detailed information on the experimental procedures, including the specific cell lines and viral strains used, the number of replicates, controls employed, and the statistical analyses performed. Describe the synthesis and purification of Bi121 and Neoilludin B in more detail, including the specific protocols followed and characterization techniques used to confirm their identity and purity.

3. Results:

The significance of the content and results is high, because the study introduces a new antiviral compound, Neoilludin B (Bi121.2), and presents its potential to combat SARS-CoV-2 and other RNA viruses. The identification of a new compound with a unique mechanism of action adds to the current body of knowledge on antiviral therapeutics.

However, there are certain issues that need to be addressed,

First, the sample size analyzed (N=2). Were the experiments performed in duplicate or triplicate replicates?

A classic experiment to determine whether the viral entry process is involved in the antiviral effect is the use of acid glycine (https://doi.org/10.1006/viro.1999.9633).

Did you perform any approach to ensure that this viral replication event is affected by pretreatment with Bi121.2?

The presentation of the results could be improved by providing more context for each finding. It would be beneficial to include a brief summary of the rationale for each experiment before presenting the results.

The statistical analyses performed should be explained more clearly. Please provide details on the specific tests used and the criteria for determining significance.

The validation of the in-silico studies and the computational model used to determine the mechanism of action of Neoilludin B is crucial. Elaborate on the methods employed to validate the predictions and the reliability of the model.

4. Discussion:

The implications of your findings should be discussed more comprehensively. Discuss the potential limitations of your study and provide suggestions for future research to address these limitations.

The synergistic effects of Neoilludin B with other anti-SARS-CoV-2 treatments, such as Molnupiravir and Nirmatrelvir, need to be substantiated with experimental evidence. Please provide data to support this claim or discuss it as a potential avenue for future research.

In conclusion, the study has the potential to contribute valuable insights to the field of antiviral research. However, significant revisions are necessary to address the methodological and presentation concerns identified in this review. I look forward to reviewing a revised version of the manuscript.

The manuscript demonstrates a reasonable command of the English language; however, there are areas that require improvement to enhance clarity and readability. Some sentences are difficult to understand and may lead to confusion for the reader. To address these issues, I recommend the following:

1. Extensive editing of English language required: The manuscript would benefit from a thorough revision to improve grammar, syntax, and sentence structure. Consider seeking assistance from a professional editor or a native English speaker to ensure that the text is clear and comprehensible.

2. Consistency in terminology and style: Ensure that the terminology and style are consistent throughout the manuscript. 

3. Proper use of abbreviations and acronyms: Introduce abbreviations and acronyms in their first occurrence, and consistently use them throughout the manuscript. Additionally, avoid overusing abbreviations, which can hinder the readability of the text.

By addressing these language-related issues, the manuscript will become more accessible to a wider audience and allow readers to better understand the content and implications of your research.

Reviewer 3 Report

The manuscript entitled Significant Broad Spectrum Antiviral activity of Bi121 Against Different Variants of SARS-CoV-2 represents an excellent piece of wrk by Subhadra et. al. The authors have shown the CC50 and ID50 of the Bi121 against Washington, Delta and Omicron variant by qPCR and anti-viral assay. I have few minor concerns related to the manuscript that must be answered.

1.  The author did not mention the Omicron strain used in the study. Did it BA.1 or BA.2 or BA.4/5. The authors mentioned that Omicron strain was isolated from South Karnataka region, India and was confirmed by S gene target failure PCR. Did the authors sequenced the viral strain and submitted the sequence to GenBank or GI?

2. What was the rationale of using E6 or HEK-ACE2 cells? The authors can choose the more relevant lung cells such as A549 cells and specialized cells for COVID-19 research A549-ACE2 cells or VERO E6+ TMPRSS2+ Ace 2 cells for stable infection as Vero E6 cells are more prone to mutations.

3. Did the authors tested this compound on other coronavirus family member like MERS-CoV or SARS-CoV-1 or human coronaviruses like NL-63? In order to claim the compound as broad spectrum antiviral.

4. the authors claims that Bi121 Interfere with Early Stages of SARS-CoV-2 Infection, Do they perform any Western blotting assay with the lysate to confirm this phenomenon? What's the mechanism behind Bi121 mediated protection.

5. Did the authors performed any SPR assay to check the affinity between the compound and virus?

6. This question is beyond the scope of the manuscript-Whether the authors performed In-vivo experiment yet because this manuscript is circulating in different journals from January 2022?
